# Impact on Potable Water Consumption Due to Massive Migrations: The Case of Bogotá, Colombia

**Nestor Ortiz** [1], **Laura González** [1] and **Juan Saldarriaga** [1,2,*]

1 Water Distribution and Sewerage Systems Research Center, Engineering School, Universidad de los Andes, Bogotá 111711, Colombia
2 Civil and Environmental Engineering Department, Universidad de los Andes, Bogotá 111711, Colombia
* Correspondence: jsaldarr@uniandes.edu.co

**Abstract:** One of the most important aspects for the regulation of a water distribution system in a city is the accurate projection of the population to be supplied. Accordingly, it is necessary to know the social, cultural, and economic characteristics of the inhabitants, as well as the industrial, commercial, tourist and migratory dynamics to understand a city's future development. Generally, population growth is continuous and usually follows a trend that makes it possible to extrapolate the future population, thus predicting the demand for drinking water based on per capita supply. However, forced migrations challenge this assumption. The present work aims to collect and analyze information related to Venezuelan migration, in the southern zone of Bogotá, Colombia. First of all, in this project, the demographic characteristics of the migrant population are defined, and the different sources of information collected during the study are used to estimate the total number of migrants in Bogotá. Then, new methodologies for tracking people using social media data connections, geolocation from active connections, and emerging platforms are shown. This is followed by an explanation of two approaches used to estimate the number of migrants from 2014 to 2020 in the boroughs of: Kennedy, Bosa, Tunjuelito, Usme, Ciudad Bolivar and the municipality of Soacha. Finally, using mathematical and statistical approximations, the study shows that the increases in water consumption in the city coincides with the flow rates needed to supply the migrant population. The results indicate that during the first half of 2021 the flow demanded by the migrant population in the studied localities was around 8 Mgal/d (350 L/s), which is 35.1% of the total flow demanded by the entire migrant population in Bogotá. The migrant water consumption was between 4.8% and 6.1% of the total demand for the city.

**Keywords:** drinking water consumption; WDS; migrant population; Bogotá—Colombia

## 1. Introduction

Demographic changes condition the adequate quantity and quality of water to be supplied to the population. These changes occur as a function of social, economic, political, and other dynamics. An accurate population projection is essential to reduce water supply vulnerability in the face of an unexpected event, yet it is complex to predict all possible scenarios. Likewise, legislation establishes both the methodology for calculating future population and the endowment with which the municipalities and/or service providers must calculate the demand for drinking water. In this regard, it is important to note that calculated drinking water supply must surpass actual drinking water demand by at least 15% [1], since new supply projects take years to design, finance and build.

On the other hand, migration crises management worldwide has been one of the most difficult political phenomena to be dealt with in recent years. A study with immigration and emigration data from19 European countries between 2002 and 2007 was conducted, to generate international migration flows that can be used to understand recent changes in migration patterns [2]; however, they mention that emigration figures reported by source

countries tend to differ from the corresponding immigration figures reported by destination countries. Hence, to overcome this obstacle, access to data must be robust and up-to-date.

According to the United Nations High Commissioner for Refugees [3] Colombia is currently the main host of Venezuelan migrants in Latin America and the second country receiving the most migrants after Turkey, due to the Syrian migration crisis. Since 2014, migration from Venezuela has been the largest human mobilization in recent history in Latin America, as the search for opportunities and better living conditions have driven the Venezuelan population to migrate from their country, be it for voluntary or forced reasons. The temporariness in the national territory ranges from the transit of people to the definitive decision to remain in Colombia for an extended period of time.

Venezuelan human mobility report [4], shows the highest percentage of Venezuelans go to Colombia mainly due to its proximity and the border dynamics between the two countries. Furthermore, the massive arrival of migrants is not exclusive to border municipalities. In fact, Cúcuta, Barranquilla, Medellín and Bogotá are the cities that have received the most Venezuelan citizens and only the first one is a border city. The capital city, Bogotá, is where most of them have settled down both formally and informally. For this reason, this study proposes the collection of migratory information related to the changes produced, specifically in the south of Bogotá. Hence, it is necessary to characterize the migrant population in the city. Subsequently, the drinking water consumption of this population is formulated using updated bibliographic sources and the user base of Bogotá's water utility (Empresa de Acueducto y Alcantarillado de Bogotá—EAAB).

As a result, the geospatial distribution of the city's migrant population is identified, thereby discretizing the average consumption by social strata, locality, hydraulic sector, and service area. This will numerically quantify the impacts on drinking water consumption and the demand absorbed by the main network because of the migratory exodus. The joint implementation of Geographic Information Systems (GIS), macro-meter readings and information from the EAAB user database, together with new sources of demographic characterization, enables us to demonstrate mathematically and statistically that the sudden increase in population in receiving areas is increasing drinking water and basic sanitation services demand in Bogotá [5], creating operational and commercial problems.

Due to this large demographic increase resulting from the Venezuelan migration, it is very important for the Colombian authorities and Bogotá's water utility to be able to supply potable water in sufficient quantity and proper quality to all the citizens. This includes other water uses like water suitcase [6]. The document provides valuable information on how population growth estimates can be generated to forecast consumption changes in the water distribution systems using novel methodologies to track people. These technologies are having a great impact, using social media data connections, geolocation from active connections and emerging platforms.

## 2. Case Study

Bogotá is located on the Andes central highlands with a population of about 7 million. It is divided into 20 districts or localities, with about 1900 total neighborhoods in the urban area. Based on information from the Spatial Data Infrastructure for the Capital District—IDECA [7] it was possible to extract the localities' map to be used in this research. The present study focuses on the localities of Bosa, Ciudad Bolivar, Kennedy, Tunjuelito, Usme and Socha. The potable water distribution network of Bogotá is composed of interconnected network systems that are used to distribute drinking water. The city has five (5) management zones and 42 hydraulic sectors. Figure 1 shows the maps of Bogotá and Soacha; in addition, the locations on which this research is focused are detailed. Through the Planning and Control Division of the EAAB, the flow macro-measurement data was collected in each part of the city, as well as the water consumption every 30 min, to have a better precision in their analysis.

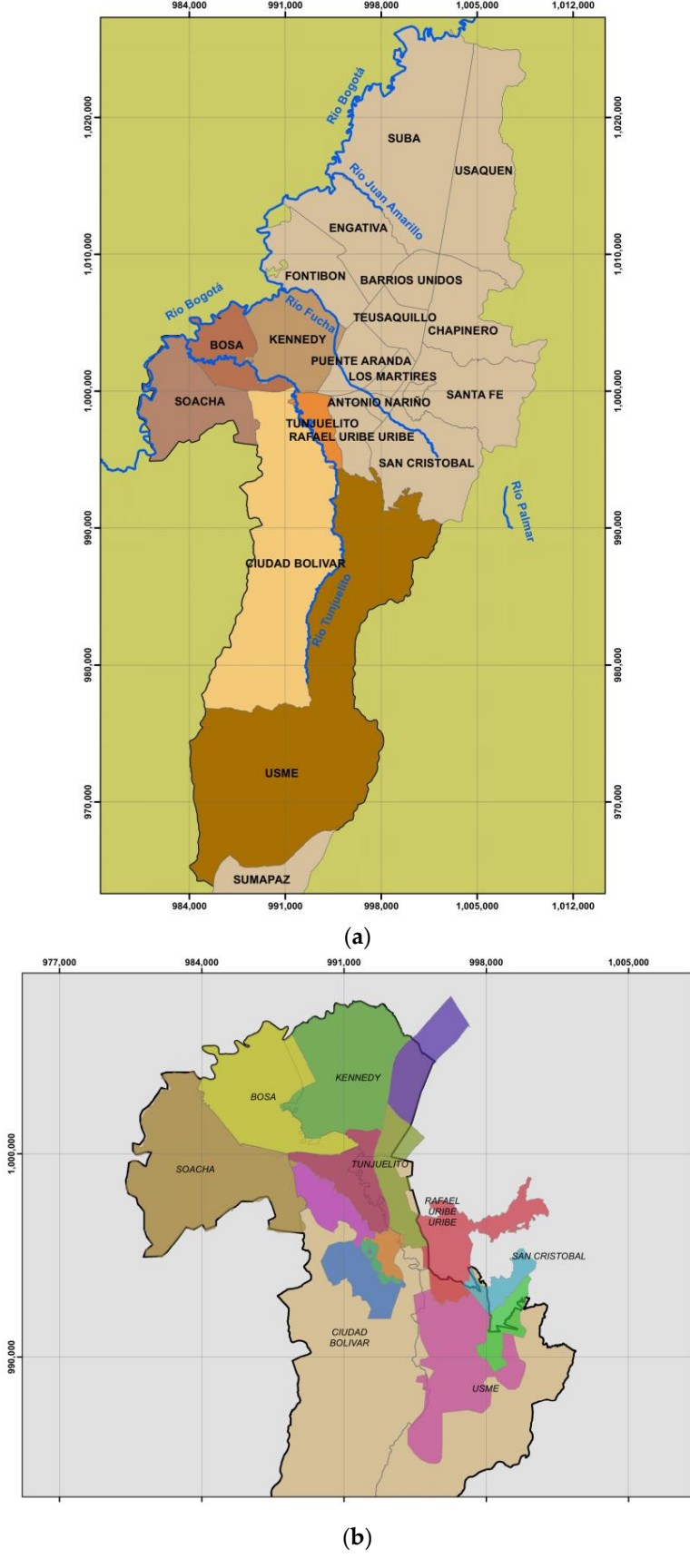

**Figure 1.** (**a**) Map of Bogotá by localities and Soacha; (**b**) Hydraulic sectors and localities to be analysed.

## 3. Materials and Methods

### 3.1. Quantifying Migratory Mobility

Besides both births and deaths, migration is one of the three most important factors in understanding the changes in size and structure of the population within a given territory. In the context of migration studies, the lack of temporal data on migrants limits the ability to address social challenges [8]. There are various sources of information that can provide an understanding of the population size in different contexts, with censuses being the main source used to record the number of people in a given area. Migración Colombia is the national migratory authority; it published the report titled "Evolución Crisis Migratoria con Venezuela 5 Años de Historia", which presents the total data on the number of migrants residing in Colombia from 2014 to 2020 [9]. On the other hand, the Regional Platform for Integral Coordination for Refugees and Migrants from Venezuela (R4V) was established to direct and coordinate the response towards refugees and migrants coming from Venezuela [10]. Within this platform, reports of the situation of Venezuelan migrants, especially in Latin America, are continuously made.

Figure 2 indicates the existence of a relationship between the values reported by Migración Colombia and R4V. Nonetheless, in certain months, the R4V platform over or underestimates the population coming from Venezuela to Bogotá.

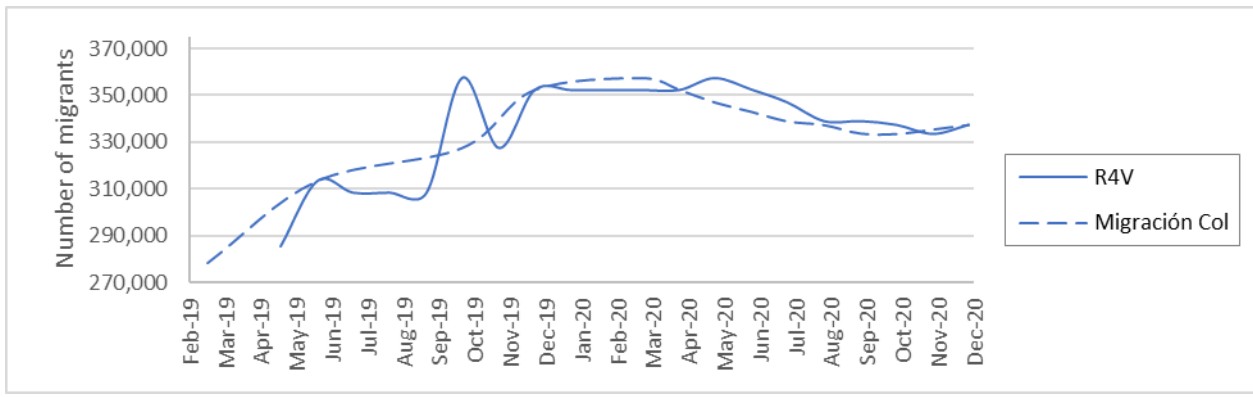

**Figure 2.** Comparison of Venezuelan migrants to Bogotá as reported by R4V vs. Migración Colombia.

Additionally, the journalistic and academic platform Venezuela Migration Project, was created by Revista Semana in 2018 to gauge, understand, and raise awareness of the migratory phenomenon. It seeks to strengthen the informative coverage and analysis of Venezuelan migration and is supported by the United States Agency for International Development (USAID) through the Partnerships for Reconciliation Program (PAR) [11].

Another platform used for the collection of demographic information is Esri Colombia, which is a leading GIS company. According to this, Figure 3 shows that the data reported by Esri is similar to the one published in the Migration Colombia reports between October 2018 and 2019. Based on this, Esri is a reliable source for obtaining an approximate number of migrants for Bogotá before October 2018 [10]. Different demographers have presented an innovative approach to estimate migrant populations by using the unexplored data source Facebook advertising platform (Application Programming Interface—marketing API), a free-access platform that allows advertisers and researchers to query sociodemographic characteristics of Facebook users [12].

Based on the Latinobarómetro report [13] about 70% of the Venezuelan migrant population uses Facebook. Similarly, the international non-profit organization iMMAP provides specialized information and support services in emergency situations, especially in humanitarian contexts. iMMAP follows migrants and refugees from Venezuela by tracking the connections of users who used to live in Venezuela and now live abroad [14]. This new methodology is being widely implemented by different demographers and scientists with applications in various research areas, particularly in social and humanitarian fields [15–21],

because a novel use of geo-tagged social media data for exploring unanswered questions related to migration theory has been demonstrated. In particular, refs. [15–17] use Facebook data, as they suggest that the approach could be successfully used to sample smaller, more dispersed groups of migrants in a given country, which are particularly difficult to reach using more traditional approaches. Thus, this represents an advantage to get information about disperse population like the one on this research. This is the first time this methodology has been applied to the drinking water sector, where migrants are tracked at a locality level in Bogotá.

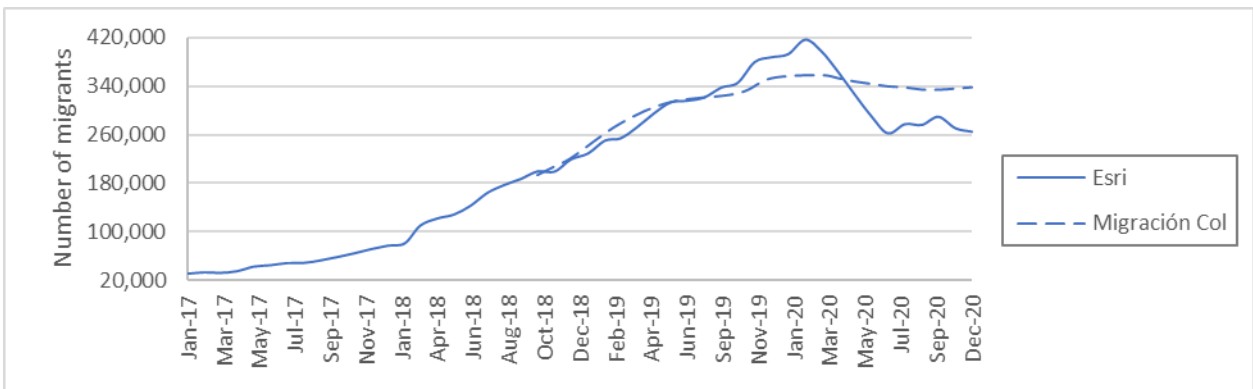

**Figure 3.** Comparison of Venezuelan migrants to Bogotá as reported by Esri vs Migración Colombia.

Figure 4 presents the monthly average number of users for Bogotá and Soacha, taking into consideration data that is published every two weeks. From this it is evident that Venezuelan users connected to Facebook from September 2018 to March 2019 were higher than the number of migrants initially reported by Migración Colombia.

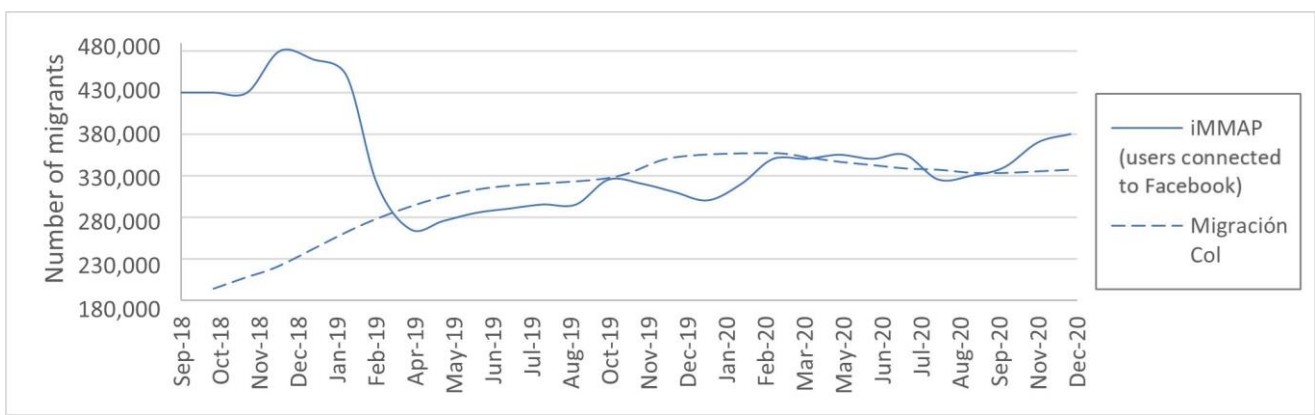

**Figure 4.** Comparison of Venezuelan migrants to Bogotá as reported by iMMAP vs. Migración Colombia.

Additionally, Tablaeu is an interactive migration data visualization platform that Migración Colombia has made available for citizens to consult, where statistics and data on Colombian and foreign migration flows can be viewed, based on the records kept by this authority at the Immigration Control Posts (PCM) in the country.

*3.2. Bayesian Method*

Bayesian statistics are based on subjective probability methods in demographic matters, i.e., they are applied to determine a specific characteristic of a group of individuals considering previous estimates that allow to update the evidence of the variable to be determined, in this case, the number of migrants per locality.

To implement this methodology, it was necessary to have various sources of information that could be compiled to obtain the most accurate estimates for the number of

migrants. According to [12] Bayesian models are particularly useful for dealing with the harmonization and clustering of demographic information in space and time. Likewise, Bayesian model is used for harmonizing and correcting the inadequacies in the available data and for estimating completely missing flows, where harmonizing refers to the process of reconciling the differences between various measurements of migration data [22]. For this reason, the statistical approach implemented by [23] uses the Metropolis-Hastings algorithm, which applies Markov Chain Monte Carlo (MCMC) methods, generating random samples based on a given probability function.

The purpose of applying Bayesian Methods sought to provide an estimate of the migrant population by location based on a combination of data from a "mathematical approximation" and the Facebook platform, such that there were two approaches to calculate the number of migrants to reduce bias in the estimations with which the estimates of drinking water demands were calculated.

Regarding the goodness-of-fit measurements, Adjusted R-squared, Pearson r correlation, Mean Absolute Error-MAPE, and Symmetric Mean Absolute Percentage Error-SMAPE were implemented to evaluate the statistical estimates [12,24,25].

### 3.3. Mathematical Approach

After compiling all the sources of information needed for this research, it was essential to connect all the databases to generate a characterization of the total number of migrants in Bogotá from 2014 to 2020. Considering the records of monthly active user connections (MAU) on Facebook between February 2019 and December 2020, a direct proportionality relationship was established between the proposed characterization regarding the number of migrants and connections according to the methodology of [19,26] and suggested by the staff of iMMAP Colombia. During the period for which there was no record of connections, extrapolations were made using the mean values to generate a historical trend of migrant behavior in each of the localities being studied.

For this method, the results are based on reported supplies, for the values of [27] in relation to the economic strata of the multiple linear regression model ("MRLM"), the values of [28] for the economic strata ("NS-031"), provisions based on EAAB users and actual provisions by [29] according to the predominant supply zone of the locality ("DR"). The totalized flow rates for the analyzed sources are shown at the end of each location.

### 3.4. Statistical Approach

With the results from the first phase, mathematical evidence on the number of migrants by localities in Bogotá was obtained, which provided the basis for the application of the Bayesian Hierarchical Method as proposed by different demographers [12,26,30,31]. To begin the statistical approach, a linear regression model was proposed as an input for the Bayesian Method to determine parameters that would reduce the squared error (SSE). This was done for the monthly total connections by locality, as shown below:

$$Y_{i,j}^{theory} = A_j * x_{i,j} + B_j \tag{1}$$

where $Y$ = number of migrants; $i$ = locality of Bogotá; $j$ = month in which the analysis is done; $x$ = monthly active connections (MAU); $A_j$ = migrant/connection; and $B_j$ = number of migrants without a Facebook account. Then, the number of migrants is determined for all localities:

$$Y_j^{theory} = \sum_n^{i=0} Y_{i,j}^{theory} \tag{2}$$

Equation (1) is substituted from the demographic data collected as shown in Equation (3):

$$Y_j^{theory} = \sum_n^{i=0} \left( A_j * x_{i,j} + B_j \right) \tag{3}$$

After determining the sum of the number of migrants, the squared error is calculated as follows:

$$SSE = \left( Y_{i,j}^{theory} - Y_{i,j}^{real} \right)^2 \tag{4}$$

Then, the Metropolis-Hastings algorithm was implemented to generate random samples based on a probability function given by the user. Furthermore, using the data obtained from the Facebook connections, the piece-wise (non-normalized) probability density function was constructed, fitting each four pieces of data to a cubic curve ("cubic spline") in order to reduce bias by increasing the polynomial degree. Subsequently, the algorithm generated about 500,000 samples of the dates for which there were records using the constructed probability function. The algorithm then generates a random sample for each point by several steps. First, the user selects a given date $x_0$. Then, a new date is calculated with the following expression:

$$x_{new} = x_0 + \eta \Delta x \tag{5}$$

where $\eta$ = a random number between $-1$ and $1$; $\Delta x$ = time step defined by the user. After, the value of the function value at both $x_{new}$ and $x_0$ and divide them is calculated, such that:

$$\alpha = \frac{f(x_{new})}{f(x_0)} \text{ if } \alpha > 1 \text{ accept } x_{new} \text{ and store it in a list ("array")}$$
$$\text{The next } x_0 \text{ is now } x_{new}$$

otherwise, the algorithm generates a new random number $\beta$ between 0 and 1:

$$\text{If } \alpha > \beta \text{ accept } x_{new} \text{ and store it in the list}$$
$$\text{If } \alpha < \beta \text{ reject } x_{new} \text{ and store } x_0 \text{ in the list}$$

The next step is to assign each of the dates in the list to its respective month. The probability of a Venezuelan with vocation to stay in a certain month will be the number of samples generated for a month divided by the total number of samples. The number of migrants in a given month is given by:

$$V_{(i,j)} = P_{(i,j)} FT_j A_{(i,j)} \tag{6}$$

where $V_{(i,j)}$ = number of Venezuelan migrants on month $i$ and locality $j$; $P_{(i,j)}$ = probability previously determined; $FT_j$ = total number of migrants that logged onto Facebook from locality $j$; and $A_{(i,j)}$ = proportionality constant calculated based on the mathematical approach.

$$A_{(i,j)} = \frac{Data_{Facebook}}{Data_{Linear\ regression}} \tag{7}$$

The last equation presents both the relationship of the data taken from Facebook and the linear regression performed.

## 4. Results and Discussions

First, drinking water consumption in Bogotá's main network was compiled based on the data provided by the trunk network management of Bogotá's water authority EAAB, as shown in Figure 5. From 2014 to 2020, the main water network experienced an increase in drinking water demand, specifically for water service zones 4 and 5, since on average the consumption of mean flows on the main network in these zones was 20.69% and 20.22%, while the other zones that make up the main network system of Bogotá consumed the remaining 59.09%. In other words, the areas considered in this study have demanded more than 40% of the total flow delivered, with the localities of Tunjuelito, Ciudad Bolívar, Usme, Bosa, Kennedy and the municipality of Soacha, contributing to a large share of this consumption.

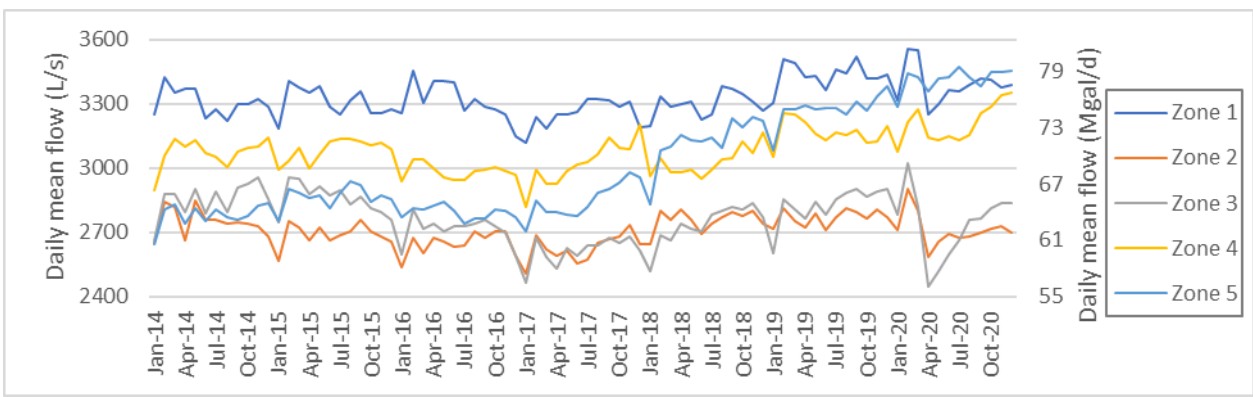

**Figure 5.** Monthly daily mean flow for supply zones in Bogotá.

Figure 6 presents the results of the proposed historical characterization for the Venezuelan migrant population in Bogotá over analysis period using Equation (5). The sources of information were grouped, and the initial input was used for the mathematical approach to estimate the number of migrants by locality.

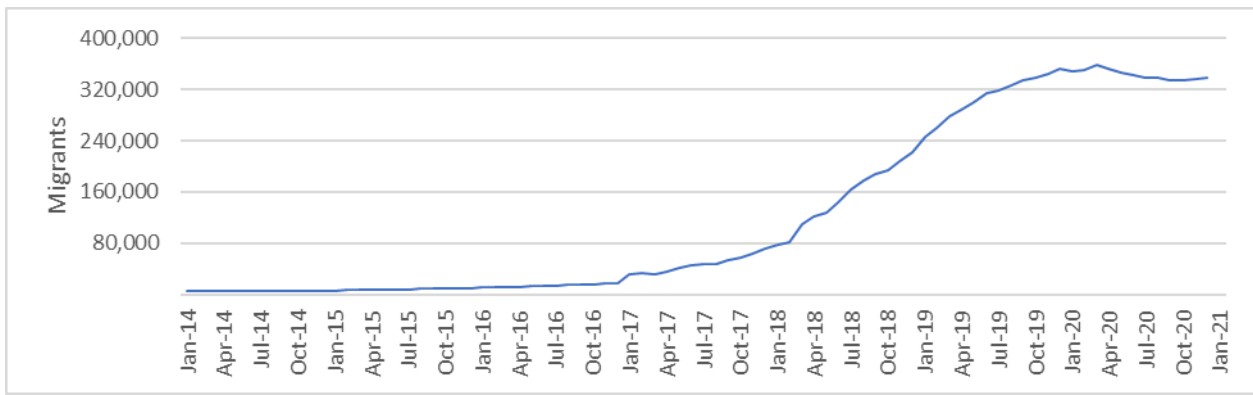

**Figure 6.** Characterization of migrant population in Bogotá.

From the proposed characterization, it can be affirmed that the data is completely in line with what is reported in the various sources of information, which means the resulting data set is congruent with what is reported in literature on migration in the country. This result is used to assign the number of users connected to Facebook by localities for the mathematical approach and provides a limit to the number of migrants, which cannot exceed these values.

Likewise, the number of monthly active users (MAU) was obtained starting from the period for which there is a record of such connections, i.e., from February 2019 to December 2020. Based on this relationship, it is possible to match the Facebook connections to approximate values of migrants per locality. Although there is still bias present in the results, it can be affirmed that, after making several approximations of the number of migrants per locality, these values are the closest to the distribution of migrants in Bogotá. The result includes Soacha, which was given a higher weighting in the Migration Colombia reports, as shown in Figure 7.

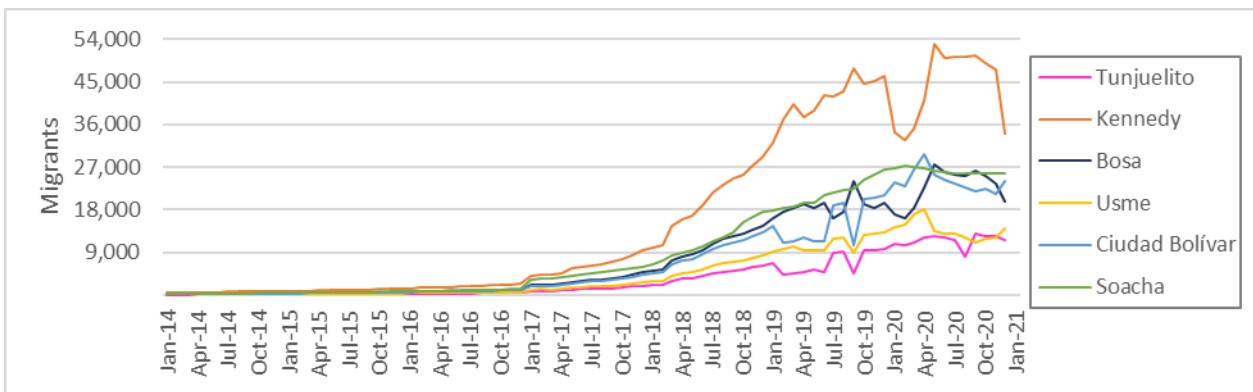

**Figure 7.** Historical estimate for number of migrants obtained by the mathematical approach for the localities being studied.

With the results for the number of migrants obtained by the statistical approximation for the period where the records of connections were available, an extrapolation of the historical data was carried out to have all the data for the analysis period. Figure 8 shows the number of migrants as estimated by this second methodology. The localities of Kennedy and Usme show the greatest changes between January 2019 and April 2020; however, the trend is similar.

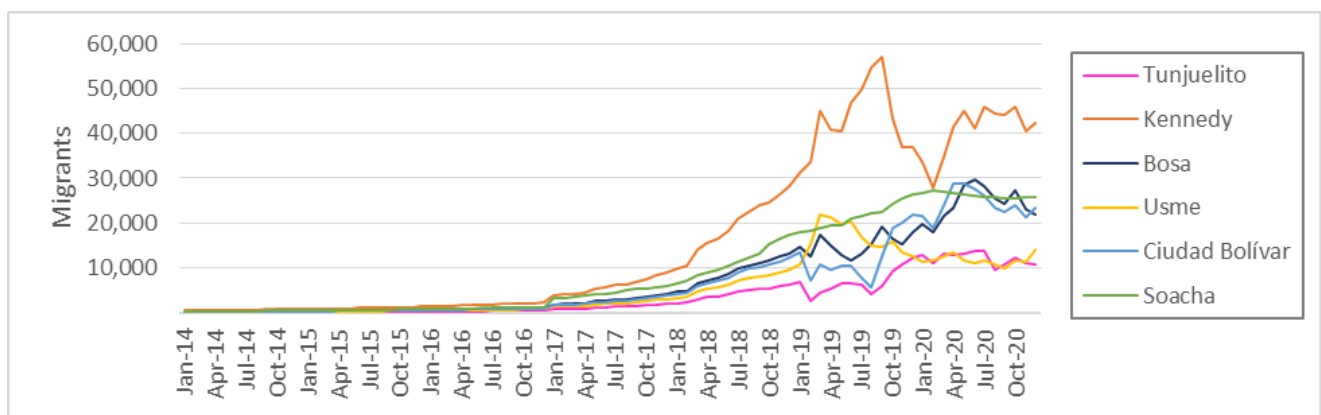

**Figure 8.** Historical estimate for number of migrants obtained by the statistical approach for the localities being studied.

To summarize, the flows contributed by the migrant population which the network had to absorb in zones 4 and 5 (Figure 9) of the water supply system are presented for the period when Bogotá received the most migrants, that is, from 2017 to 2020, and the two approaches are grouped and compared with the total flows reported for the study zones. In the determination of the daily mean flow using the suggested supplies from NS-03, the results were always lower compared to the other three adopted sources. For zone 4 the flow rates were higher implementing the user base; also, in zone 5 the MRLM and user base flow rates were higher.

Moreover, the demand consumed by migrants is below 10% of the monthly daily mean flow demanded and the trend of the two approximations remains almost constant. The peak migrant flow is defined by the user EAAB, with the period between March and June 2020 being the highest and with an evident decrease throughout the first year of the pandemic.

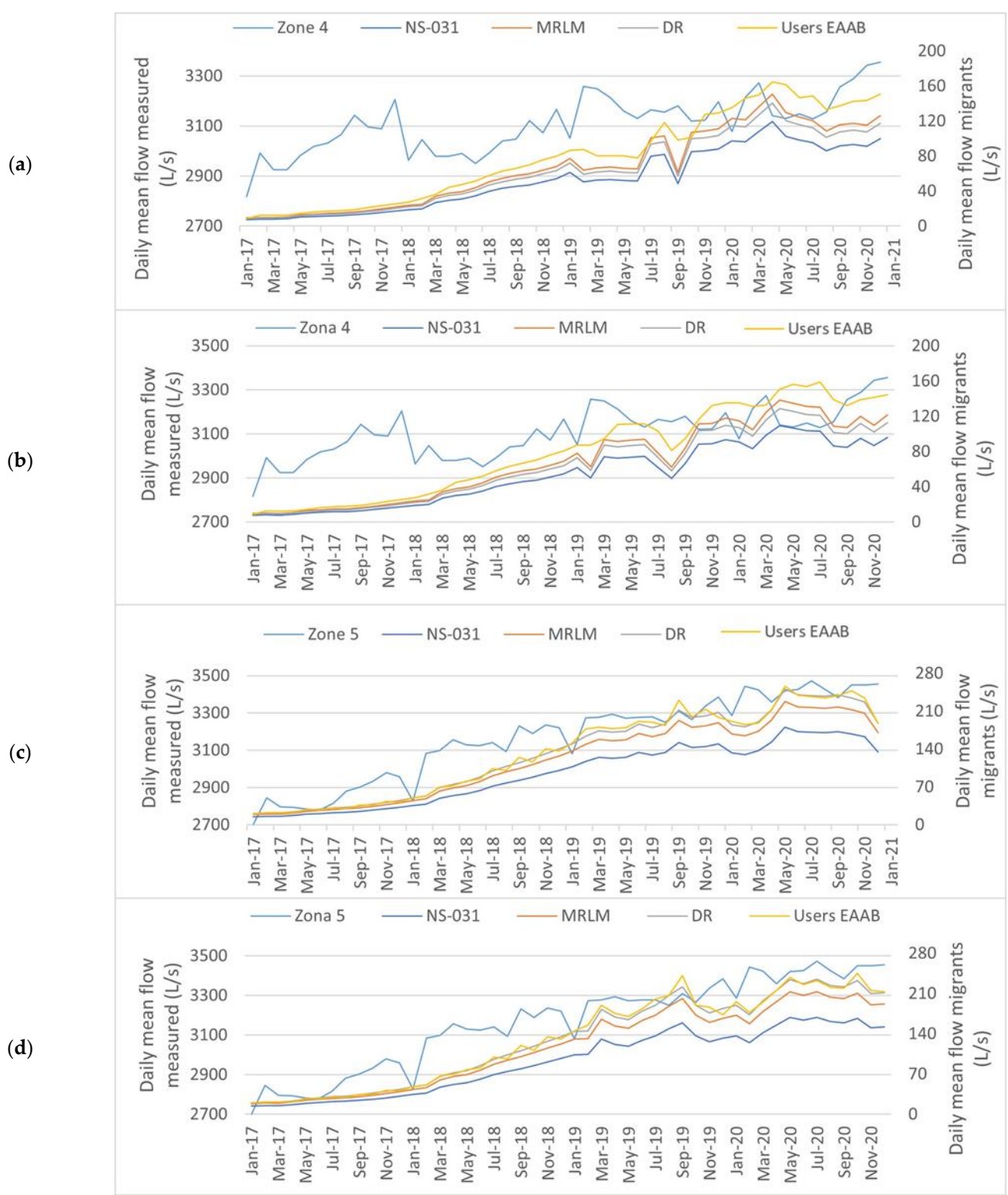

**Figure 9.** Monthly daily mean flow left and migrant consumption right (**a**) Zone 4 by the mathematical approach; (**b**) Zone 4 by the statistical approach; (**c**) Zone 5 by the mathematical approach; (**d**) Zone 5 by the statistical approach.

Lastly, Figure 10 present the totalized flow rates obtained using both methodologies for all four sources of information. The migration of people from Venezuela has had a direct impact on the demands for drinking water. This research presents one of the reasons that affected the operation within the main network, specifically in the southern areas of the city

of Bogotá, from 2014 to 2020. Despite the difference in flow rate values, the behavior of the data is similar, which makes it feasible to use either of the two methodologies, mathematical and statistical.

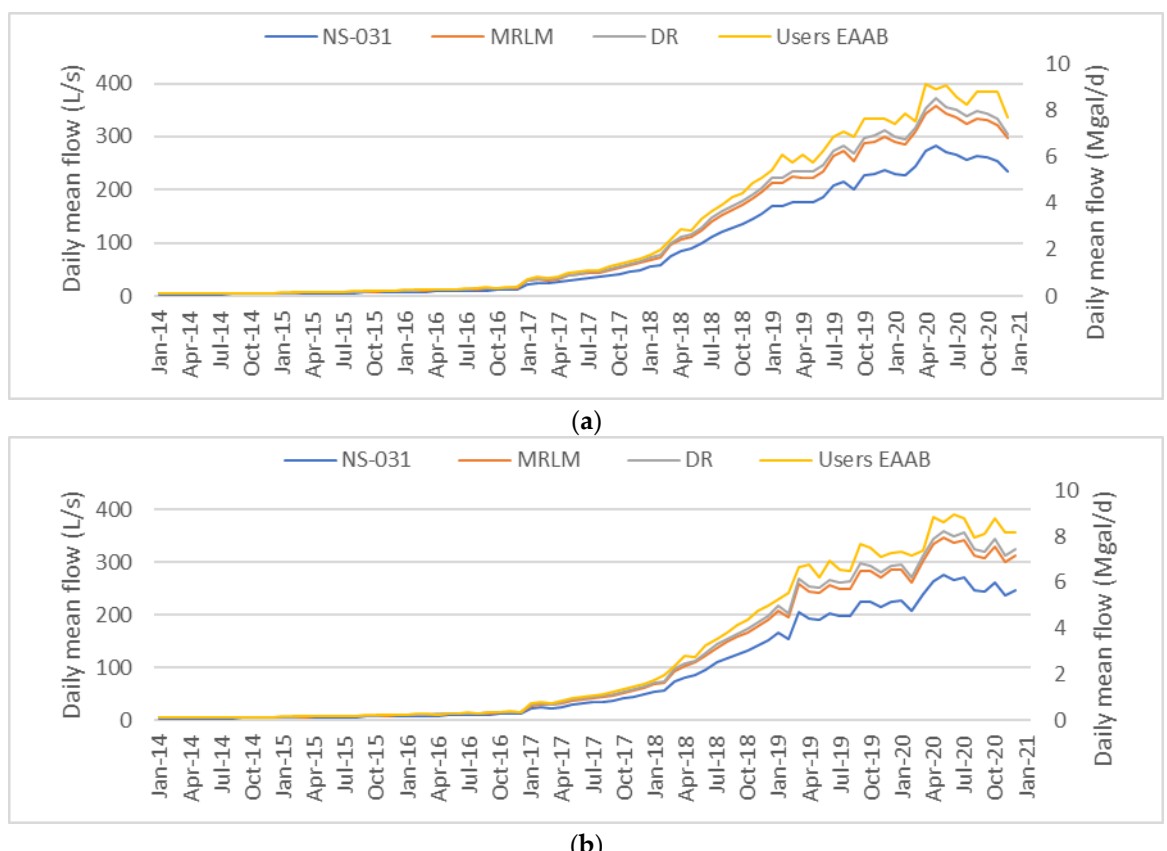

**Figure 10.** Total daily mean flow for localities being studied (**a**) Mathematical approach; (**b**) Statistical approach.

## 5. Conclusions

According to the results presented in this study, there has been a direct, significant impact on drinking water consumption due to the massive reception of migrants to Bogotá. The dynamics that this population has adopted stand out since approximately 20% of the total number of migrants arriving in Colombia are in Bogotá. Likewise, 49.26% were in a regular migratory situation, while the other 50.74% were in an irregular migratory situation, highlighting that locating this population is paramount, a factor that non-traditional sources (Facebook API) do not restrict with respect to the limitations imposed by traditional censuses.

A historical estimation for the number of migrants that Bogotá has received was made by means of different sources of information, prioritizing the information reported by the migratory authority. Regarding the number of migrants that have arrived in the city, it is possible to affirm that Suba, Kennedy, Ciudad Bolívar, Bosa, Engativá and Soacha (municipality) have the highest number of migrants, since the segregation presented by the population coming from Venezuela is like the local context, where 81% of the migrants are located in low income (strata 2 and 3), compared to 74.8% in the local context.

Overall, based on the results obtained for the locations studied, both the historical curves for measured daily mean flow and the curves for the daily mean flow of the migrant population were generated, showing that the increases in water consumption coincide with the estimated flow rates for the migrant population in zones 4 and 5.

There has been a chain migration in Colombia since 2014. For the case of Bogotá, in March 2020 there was a migration peak with approximately 385,000 migrants, including Soacha, which was attenuated due to the pandemic.

By 2021, the figures have not yet returned to the exponential historical tendency that had been occurring since 2017–2018 due to the phenomenon of returning migration; however, with the implementation of the Temporary Protection Statute for Venezuelan Migrants (ETPV), which is a complementary mechanism to the international refugee protection regime, there is still uncertainty as to whether there will be a gradual return of migrants, a circumstance that the methodology adopted in this research is not yet able to predict.

When compiling the total flows by migrant population based on the two adopted methodologies, both results indicate that during the first half of 2021 the flow demanded by the migrant population in the studied localities was around 8 Mgal/d, which is 35.1% of the flow demanded by the entire migrant population in Bogotá. The migrant water consumption was between 4.8% and 6.1% of the total demand for the city.

One should mention that, in Bogotá, the local and migrant population is implementing water collection and reutilization habits, as well as the implementation of water-saving sanitary devices that have a positive impact on the demand and on the reduction of water flows. Nonetheless, the focus of this study shows that during peak migration period, that is, during the first half of 2020, the demand increased by 1.90% of the total consumption supplied by EAAB.

This first insight into the water consumption of a population outside a particular city using new technologies and taking advantage of the ongoing digital transformation, is a tool that urban drinking water distribution system managers should implement for their decision making and real-time planning of hydraulic systems.

The implemented methodologies reduce the bias that non-traditional sources have for tracking people; thus, bringing it closer to the concept of a smart city. In addition, processing information about human mobility phenomena within the city through permanent sensors (cell phone connections and social networks) is an important requirement to generate a Digital Twin. In contrast, water utilities should also implement tools to collect data in real time, so that decision makers in SCADA systems can quickly adapt to changes in demand and, if necessary, perform simulations to generate scenarios to address any challenge within the system, since the lack of data reduces the accuracy of the results.

**Author Contributions:** Conceptualization, N.O. and J.S.; methodology, N.O.; software, N.O.; validation, N.O. and J.S.; formal analysis, N.O. and J.S.; investigation, N.O., L.G. and J.S.; data curation, N.O.; writing—original draft preparation, N.O. and L.G.; writing—review and editing, L.G. and J.S.; visualization, N.O. and L.G.; supervision, L.G. and J.S.; project administration, J.S. All authors have read and agreed to the published version of the manuscript.

**Funding:** This research received no external funding.

**Informed Consent Statement:** Not applicable.

**Data Availability Statement:** The data presented in this study are available on request from the corresponding author. The data are not publicly available due to privacy issues from water utilities.

**Acknowledgments:** The authors would like to thank Empresa de Acueducto y Alcantarillado de Bogotá (EAAB), iMMAP Colombia, Migración Colombia, and DANE for providing access to water use data.

**Conflicts of Interest:** The authors declare no conflict of interest.

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
