# Peer review of "Impact on Potable Water Consumption Due to Massive Migrations: The Case of Bogotá, Colombia"

_water, doi:10.3390/w14243987_

Round 1

Reviewer 1 Report

Dear authors,

the article deals with an important topic of the impact of migration on water consumption in Bogota. The work is an interesting approach to the subject. However, I have a few comments on the publication, which I include below:

1. In the Introduction section, you mention that "In this regard, it is important to note that calculated drinking water supply must surpass actual drinking water demand by at least 15%, since new supply projects take years to design, finance and build". Please add missing reference to this statement.

2. The work lacks reference to research that has been conducted around the world on the impact of migration on potable water consumption. Please complete the literature review on how this topic has been addressed in other countries.

3. In Methodology section, you mention: "This new methodology is being widely implemented by different demographers and scientists with applications in various research areas, particularly in social and humanitarian fields [1218]". Reference to 7 sources without justification seems too general. Referring to sources requires specifying the context in which the research presented in the literature relates to the subject of the work. Please, extend the description of cited works.

Reviewer 2 Report

The review topic is really interesting and the manuscript is well-written. Therefore, the manuscript has some problems that are listed below:

1) Lines 17-19. The authors mentioned all in the future tense. Is this content discussed in the manuscript? Or planned in the next study? Please, clarify it.

2) The author should highlight more results in the abstract. Was a new method used in the manuscript? The authors should highlight it.

3) The presentation of the figures should be improved to sound more scientific.

4) Figure 4 is too confusing. Please explain in the figure caption the axis x and y and the way appropriate to understand the data.

5) Why was the Bayesian Method used? Please, explain it.

6) The authors stated that migration had a direct impact on drinking water consumption. Please, add the procedures the authorities did to deliver water to all the population during this period. Do the authorities plan some actions for the new future? How can your study contribute to those actions?

Reviewer 3 Report

Impact on potable water consumption due to massive migrations. The case of Bogotá, Colombia:

·        Add some of the most important quantitative results to the Abstract.

·        Add/Replace the name of the study area with the Keywords.

·        In the last paragraph of the Introduction, the authors should mention the weak point of former works (identification of the gaps) and describe the novelties of the current investigation to justify that the paper deserves to be published in this journal.

·        Discuss the most important reasons for the variations in the monthly daily mean flow left and migrant consumption.

·        Focus on the advantages/disadvantages of the proposed method concerning the obtained results.

·        How can expand the results to other regions with similar/different climates?

·        The quality of the language needs to be improved for grammatical style and word use.

Round 2

Reviewer 3 Report

Acceptable.